# The microRNA-210-Stathmin1 Axis Decreases Cell Stiffness to Facilitate the Invasiveness of Colorectal Cancer Stem Cells

**DOI:** 10.3390/cancers13081833

**Published:** 2021-04-12

**Authors:** Tsai-Tsen Liao, Wei-Chung Cheng, Chih-Yung Yang, Yin-Quan Chen, Shu-Han Su, Tzu-Yu Yeh, Hsin-Yi Lan, Chih-Chan Lee, Hung-Hsin Lin, Chun-Chi Lin, Ruey-Hwa Lu, Arthur Er-Terg Chiou, Jeng-Kai Jiang, Wei-Lun Hwang

**Affiliations:** 1Graduate Institute of Medical Science, College of Medicine, Taipei Medical University, Taipei 110, Taiwan; liaotsaitsen@tmu.edu.tw (T.-T.L.); sandylan@tmu.edu.tw (H.-Y.L.); 2Cell Physiology and Molecular Image Research Center, Wan Fang Hospital, Taipei Medical University, Taipei 116, Taiwan; 3Ph.D. Program for Cancer Molecular Biology and Drug Discovery, China Medical University, Taichung 406, Taiwan; wccheng@mail.cmu.edu.tw; 4Research Center for Cancer Biology, China Medical University, Taichung 406, Taiwan; 5Department of Education and Research, Taipei City Hospital, Taipei 106, Taiwan; A3703@tpech.gov.tw; 6General Education Center, University of Taipei, Taipei 100, Taiwan; 7Cancer Progression Research Center, National Yang Ming Chiao Tung University, Taipei 112, Taiwan; ycchen123@ym.edu.tw; 8Institution of Microbiology and Immunology, National Yang-Ming University, Taipei 112, Taiwan; d49702015@ym.edu.tw (S.-H.S.); g30102009@gm.ym.edu.tw (T.-Y.Y.); 9Department of Biotechnology and Laboratory Science in Medicine, National Yang Ming Chiao Tung University, Taipei 112, Taiwan; chihchanlee@gmail.com; 10Department of Biotechnology and Laboratory Science in Medicine, National Yang-Ming University, Taipei 112, Taiwan; 11Institute of Clinical Medicine, National Yang Ming Chiao Tung University, Taipei 112, Taiwan; hhhlin7@vghtpe.gov.tw; 12Division of Colon & Rectal Surgery, Department of Surgery, Taipei Veterans General Hospital, Taipei 112, Taiwan; cclin15@vghtpe.gov.tw; 13School of Medicine, National Yang Ming Chiao Tung University, Taipei 112, Taiwan; 14Department of Surgery, Zhongxing Branch, Taipei City Hospital, Taipei 106, Taiwan; DAK23@tpech.gov.tw; 15Institute of Biophotonics, National Yang Ming Chiao Tung University, Taipei 112, Taiwan; aechiou@ym.edu.tw

**Keywords:** colon cancer, cancer stem cells, microRNAs, deformability

## Abstract

**Simple Summary:**

Metastasis of tumor cells is the leading cause of death in cancer patients. Concurrent therapy with surgical removal of primary and metastatic lesions is the main approach for cancer therapy. Currently, therapeutic resistant properties of cancer stem cells (CSCs) are known to drive malignant cancer progression, including metastasis. Our study aimed to identify molecular tools dedicated to the detection and treatment of CSCs. We confirmed that microRNA-210-3p (miR-210) was upregulated in colorectal stem-like cancer cells, which targeted stathmin1 (STMN1), to decrease cell elasticity for increasing mobility. We envision that strategies for softening cellular elasticity will reduce the onset of CSC-orientated metastasis.

**Abstract:**

Cell migration is critical for regional dissemination and distal metastasis of cancer cells, which remain the major causes of poor prognosis and death in patients with colorectal cancer (CRC). Although cytoskeletal dynamics and cellular deformability contribute to the migration of cancer cells and metastasis, the mechanisms governing the migratory ability of cancer stem cells (CSCs), a nongenetic source of tumor heterogeneity, are unclear. Here, we expanded colorectal CSCs (CRCSCs) as colonospheres and showed that CRCSCs exhibited higher cell motility in transwell migration assays and 3D invasion assays and greater deformability in particle tracking microrheology than did their parental CRC cells. Mechanistically, in CRCSCs, microRNA-210-3p (miR-210) targeted stathmin1 (STMN1), which is known for inducing microtubule destabilization, to decrease cell elasticity in order to facilitate cell motility without affecting the epithelial–mesenchymal transition (EMT) status. Clinically, the miR-210-STMN1 axis was activated in CRC patients with liver metastasis and correlated with a worse clinical outcome. This study elucidates a miRNA-oriented mechanism regulating the deformability of CRCSCs beyond the EMT process.

## 1. Introduction

Cytoskeletal components, including microtubules, actins, and intermediate filaments, support the structure of eukaryotic cells with appropriate viscoelasticity to regulate physiological cell morphology [1], division [2,3,4], and movement [5,6]. During cancer progression, intercellular communication and cell-extracellular matrix (ECM) interactions define the localized and premetastatic tumor microenvironment (TME) for stimulating cancer metastasis [7]. Although cellular viscoelasticity has been studied using microrheology to determine the intracellular elastic and viscous moduli [8,9,10], little is known about the viscoelasticity of cancer cells during stepwise metastatic progression in distinct TMEs.

Cancer stem cells (CSCs), a nongenetic source of phenotypic heterogeneity in bulky tumors, are responsible for tumor initiation, therapeutic resistance, and distal metastasis [11]. In colorectal cancer (CRC), CD133(+) and ESA(+)CD44(+) CSCs have been identified [12,13,14]. CD26(+) and Lgr5(+) CSCs are further suggested to contribute to the maintenance of distal metastasis [15,16]. CSC phenotypes can be induced by epithelial-mesenchymal transition (EMT) inducers [17], maintained by inflammatory cytokines/chemokines or defined by Wnt activity [18]. However, the mechanism by which CSCs modulate cellular viscoelasticity to promote local invasion and distal metastasis remains elusive.

MicroRNAs (miRNAs) are endogenously expressed small noncoding RNAs of 18–24 nucleotides in length that modulate gene expression at the posttranscriptional level [19]. Dysregulation of miRNAs contributes to tumor formation and progression [20]. Several miRNAs have been associated with EMT, angiogenesis, ECM remodeling, proliferation, invasion, and apoptosis in liver or lung metastases of CRC [21]. miR-20a-5p promotes CRC invasion and metastasis by downregulating Smad4 [22]. miR-885-5p downregulates CPEB2, a negative regulator of TWIST1, and induces cytoskeletal rearrangement by upregulating Rho family small GTPases [23]. However, the miRNome responsible for cellular viscoelasticity is undefined.

This study reveals an EMT-independent mechanism for motility control and demonstrates that modulation of colorectal cancer stem cell (CRCSC) stiffness through miR-210-3p (miR-210) and its downstream target stathmin1 (STMN1) is essential for CRCSC invasiveness. The miR-210^High^/STMN^Low^ signature is further associated with liver metastasis of CRC and predicts a worse clinical outcome.

## 2. Materials and Methods

### 2.1. Plasmids, shRNA Clones and Synthetic Oligonucleotides

The miR-210 antagomir and scramble control were purchased from RiboBio Co., Guangzhou, China. The miR-210 agomir and agomir control were synthesized by GenePharma, Shanghai, China. The miR-Zip Control and miR-Zip-210 plasmids were purchased from System Bioscience, Palo Alto, CA, USA. The pLenti and pLenti-STMN1- Myc-DDKvectors were obtained from OriGene, Rockville, MD, USA. The pCMVΔR8.9, pDVsVg, and pLKO.1-shLuc vectors and the shRNA against STMN1 were obtained from the National RNAi Core Facility of Taiwan for gene silencing. All clones were verified by direct sequencing.

### 2.2. Cell Culture

Human embryonic kidney (HEK) 293 cells were cultured in Dulbecco’s modified Eagle’s medium (Gibco/Life Technologies, New York, NY, USA). Human colorectal carcinoma HT29, HCT15 and Colo205 cells were cultured in RPMI-1640 medium (Gibco/Life Technologies). SW1116 human colorectal carcinoma cells were cultured in L-15 medium (Gibco/Life Technologies). The above media were supplemented with 10% fetal bovine serum (FBS, Gibco/Life Technologies). HEK293, HT29, HCT15 and Colo205 cells were cultured in a humidified incubator at 37 °C with 5% CO_2_. SW1116 cells were cultured in a humidified incubator at 37 °C with atmospheric air. HCT15-vec and HCT15-Snail stable clones were generated previously [24]. The authenticity of cell lines was verified by examining their DNA-short tandom repeat (STR) profiles.

### 2.3. Expansion of Colorectal Cancer Stem Cells

To expand sphere-derived cancer stem cells (SDCSCs), a single-cell suspension was prepared, and cells were cultured in stem cell medium (SCM; DMEM/F12 supplemented with N2 Plus Supplement (Invitrogen, New York, NY, USA), 10 ng mL^−1^ recombinant bFGF (PeproTech Asia, Suzhou, China), 10 ng mL^−1^ EGF (PeproTech Asia) and 1% penicillin–streptomycin (Gibco/Life Technologies) for 20 days to form tumor spheres. TryPLE express (Gibco/Life Technologies) was used to dissociate cells and SDCSCs to prepare single-cell suspensions for experiments. Cells were cultured in a humidified incubator at 37 °C with 5% CO_2_.

### 2.4. Transwell Migration Assay

RPMI medium (600 μL) supplemented with 10% FBS was added to the bottom wells, and 2 × 10^5^ cells suspended in basal RPMI medium were then seeded in the 6.5 mm diameter upper chamber with an 8 μm pore size membrane (Corning, New York, NY, USA) and incubated for 20 h. Cell suspensions in the upper inserts were discarded, and the remaining cells were removed with cotton swabs. Cells adhering to the underside of the membranes were fixed with 4% paraformaldehyde (Sigma-Aldrich, St. Louis, MO, USA) for 15 min and stained with 1% crystal violet reagent (Sigma-Aldrich) for 30 min at room temperature. Images were acquired with an inverted microscope (Eclipse Ts-2, Nikon Instruments Inc., Tokyo, Japan), and the migrated cells in a 10x low-power field (LPF) were counted for quantification.

### 2.5. Two-and-a-Half Dimentional (2.5D) Time-Lapse Trajectory

For measuring 2.5D cell motility, a mixture of 0.85 mL of 3 mg mL^−1^ PureCor bovine collagen solution (Advance Biomatrix, Carlsbad, CA, USA), 0.3 mL of 5× RPMI basal medium, 6 μL of 1 M NaOH and 0.35 mL of water to a total volume of 1.5 mL was prepared as the collagen solution. Then, 250 μL of the collagen solution was added to wells in a 24-well plate for solidification at 37 °C for 30 min. A total of 3 × 10^4^ cells were suspended in RPMI basal medium and seeded on top of the thick collagen layer for 6 h prior to time-lapse recording using an IX83 inverted microscope (Olympus Corporation, Tokyo, Japan). Images were acquired every 10 min for up to 5 h, and a video was exported using cellSens software (Olympus Corporation).

### 2.6. Three-Dimensional (3D) Invasion Assay

A total of 1 × 10^6^ cells were suspended in 500 μL of basal RPMI medium and plated in one well of a four-well chambered borosilicate coverglass slide (Lab-tek, New York, NY, USA) overnight for attachment. Then, the supernatant was removed, and the surface of the well was covered with 350 μL of the collagen solution described above for 30 min to allow solidification. Then, 700 μL of basal RPMI medium was added, and the slide was incubated at 37 °C and 5% CO_2_ for 72 h. After incubation, cells were fixed with 4% formaldehyde (Sigma-Aldrich) at 4 °C overnight and mounted with Fluoroshield with DAPI (Sigma-Aldrich). Confocal images (49 layers) of each well were acquired at 1.5 μm steps from the bottom to a height of 73.5 μm with an Olympus FV1000 laser confocal microscope (Olympus Corporation).

### 2.7. Paired Cell Assay

For BrdU labeling, cells were cultured in medium containing 0.5 μM BrdU (Sigma-Aldrich) for 2 weeks to ensure BrdU incorporation in cells. Cells were then synchronized with 40 ng mL^−1^ nocodazole (Sigma-Aldrich) overnight in the presence of 0.5 μM BrdU. For the BrdU chase, cells were washed intensively, trypsinized and seeded on poly-L lysine (Sigma-Aldrich)-coated coverslips placed in wells of 6-well plates in BrdU-free medium and synchronized through sequential exposure to thymidine, nocodazole, and blebbistatin [25] to control cell division and entry into the second mitosis. Briefly, cells were treated with 5 mM thymidine (Sigma-Aldrich) for 24 h and released for 10 h prior to additional thymidine treatment for 24 h (the first round of division). The thymidine was then removed for 6 h prior to nocodazole treatment for 16 h to enrich mitotic-phase cells. The nocodazole was then removed by washing for 15 min prior to 50 μM blebbistatin (Sigma-Aldrich) treatment for up to two hours (the second round of division). Paired cells were observed at this stage. Cells were fixed with 4% paraformaldehyde for 30 min at 4 °C and permeabilized with 0.1% Triton X-100 for 5 min. Then, cells were immersed first in 1 N HCl for 10 min on ice and then in 2 N HCl/1% Triton X-100 for 45 min in a 37 °C incubator to open the DNA helix. Immediately after one acid wash with PBS, borate buffer (0.1 M, pH = 8) was used to buffer cells for 12 min at room temperature. Cells were washed again with PBS three times and incubated overnight with an antibody against BrdU (1:200, 14-5071-82, eBioscience, San Diego, CA, USA) and a fluorescein-conjugated goat anti-mouse antibody (1:200, F2761, Invitrogen). Cells were mounted with Fluoroshield with DAPI (Sigma-Aldrich), and images were acquired with a Leica DM600B fluorescence microscope (Leica Microsystems, Wetzlar, Germany).

### 2.8. Elasticity Measurements

Video particle tracking microrheology (VPTM) was used to measure the elastic modulus of cells [4,26]. A total of 2 × 10^6^ cells were suspended in basal RPMI medium in a 10-ch dish, and 20 µL of 200 nM fluorescent carboxylated polystyrene particles (F8810, Invitrogen, fluorescence excitation/emission peaks: 580 nm/605 nm) was then injected into the cells with a biolistic particle delivery system (PDS-100; pressure, Bio-Rad, 450 psi, Hercules, CA, USA). Three hours after particle injection, the cells were washed twice with PBS and transferred to 35 mm glass bottom culture dishes (Alpha Plus, Saitama, Japan). After incubation for 4 h, the two-dimensional Brownian motion of intracellular fluorescent beads was recorded with an inverted epifluorescence microscope (Eclipse Ti, Nikon Instruments Inc.), equipped with an oil immersion objective (100×, NA = 1.45), a sCMOS camera (Hamamatsu, Hamamatsu-shi, Japan), and a cell incubation chamber (INUB-GSI-F1, TOKAI HIT, Fujinomiya City, Japan). The two-dimensional projection of the trajectories of the intracellular fluorescent beads was recorded for 10 s at a frame rate of 100 Hz and analyzed via customized MATLAB software (MathWorks, Natic, MA, USA) [27]. From the two-dimensional (2-D) position (x(t), y(t)) of each particle as a function of time, we calculated the ensemble-averaged mean square displacement (MSD, Kenilworth, NJ, USA), the effective creep compliance, and the elastic modulus G’(ω) [28]. The subcellular locations of injected particles were observed using confocal microscopy (LSM 880, ZEISS, Oberkochen, Germany), and 3D images were generated using ZEISS Zen software.

### 2.9. Cell Viability, Clonogenicity and Sphere Formation Assays

A total of 1 × 10^4^ cells were suspended in 100 μL of complete RPMI medium and seeded in wells in 96-well plates for 48 h. The medium was discarded, and MTT reagent (Sigma-Aldrich) was added to the cells for 45 min at 37 °C. Mitochondrial MTT crystals were dissolved with DMSO (J.T. Baker, Phillipsburg, NJ, USA), and the optical density values were then read in a microplate reader (SpectraMax 250, Molecular Devices Corp., San Jose, CA, USA). To evaluate clonogenicity, 1 × 10^4^ cells were resuspended in complete RPMI medium and seeded in wells of a 6-well plate for 10 days. Colonies were visualized by crystal violet staining prior to counting. For sphere formation assays, 1000 cells were suspended in SCM, and the spheroids were counted after 10 days. Images were acquired with an inverted microscope (Eclipse Ts-2, Nikon Instruments Inc.), and spheroids in a 4× LPF were counted for quantification.

### 2.10. Lentivirus Production and Transduction

For virus packaging, pCMVΔR8.9, pDVsVg and expression lentivectors (miR-Zip control, miR-Zip-210, pLenti-vector, pLenti-STMN1 Myc-DDK, pLKO.1-shLuc, and shRNA clones) were cotransfected into 293T cells with T-Pro NTR III reagent (T-Pro Biotechnology, Taiwan) overnight according to the manufacturer’s protocol. The virus-containing supernatant was harvested 48 h after transfection. Cells were seeded and supplemented with 8 μg/mL polybrene (Sigma-Aldrich) for overnight virus transduction.

### 2.11. RNA Extraction and Quantitative RT-PCR

Cells were immersed in TRIzol^®^ reagent (Life Technologies). Total RNA was reverse transcribed using a RevertAidTM Reverse transcriptase kit (Fermentas, Waltham, MA, USA), and FAST SYBR Green Master Mix (Applied Biosystems Inc., Foster City, CA, USA) was used for real-time PCR in a StepOne-Plus real-time PCR system (Applied Biosystems Inc.). Cellular mRNA and miRNA expression levels were normalized to the expression levels of GAPDH and U6, respectively. The sequences of the primers used are indicated below. Primer for reverse transcription of miR-210-3p (GTC GTA TCC AGT GCA GGG TCC GAG GTA TTC GCA CTG GAT ACG ACA CAG GC). Primers for qPCR analysis: MiR-210-3p (forward primer: GGG GGG AAT ATA ACA CAG ATG GCC, reverse primer: TGC AGG GTC CGA GGT), GAPDH (forward primer: GGA GTC CAC TGG CGT CTT CA, reverse primer: TGG TTC ACA CCC ATG ACG AA); U6 (forward primer: CGC TTC GGC AGC ACA TAT AC, reverse primer: TTC ACG AAT TTG CGT GTC AT), LRRC2 (forward primer: CTT GGC AGA AGA AGG AGG TG, reverse primer: AGT ATA CAG CCT GGG GGA TG), CDKN2A (forward primer: ACC AGA GGC AGT AAC CAT GC, reverse primer: AAG TTT CCC GAG GTT TCT CA), PARD3: (forward primer: TTT CAG CCT CAT CCA GCA G, reverse primer: TTC CTC CAT CTC CAT GTT CC), RCBTB2 (forward primer: TCG TCA GGC TTG TGT CTT TG, reverse primer: CGT CAC CTA ACC CCA AAC AG), STMN1 (forward primer: TAC ACT GCC TGT CGC TTG TC, reverse primer: AGG GCT GAG AAT CAG CTC AA), CDH1 (forward primer: AGA TGG CCT TAG AGG TGG GT, reverse primer: AGG CTG TGC CTT CCT ACA GA), CDH2 (forward primer: AGC TTC TCA CGG CAT ACA CC, reverse primer: GTG CAT GAA GGA CAG CCT CT), SNAI1 (forward primer: GCT GCC AAT GCT CAT CTG GGA CTC T, reverse primer: TTG AAG GGC TTT CGA GCC TGG AGA T), NANOG (forward primer: CAA CCA GAC CCA GAA CAT CC, reverse primer: TTC CAA AGC AGC CTC CAA G), POU5F1 (forward primer: ACC GAG TGA GAG GCA ACC, reverse primer: TGA GAA AGG AGA CCC AGC AG), LGR5 (forward primer: TGT TGG GAG ATC TGC TTT C, reverse primer: CAG ACG GTT TGA GGA AGA GA), CD44 (forward primer: CCA GAT GGA GAA AGC TCT GA, reverse primer: GTC ATA CTG GGA GGT GTT GG), VIM (forward primer: CAA TGT TAA GAT GGC CCT TG, reverse primer: GGG TAT CAA CCA GAG GGA GT), BMP4 (forward primer: CTC CTG GTC ACC TTT GGC CA, reverse primer: ATT CCA GCC CAC ATC GCT GA), and CDX2 (forward primer: CTG GAG CTG GAG AAG GAG TTT C, reverse primer: ATT TTA ACC TGC CTC TCA GAG AGC).

### 2.12. Bioinformatic Analysis

The small RNA-seq (smRNA-seq) data of HT29 cells, HCT15 cells and expanded SDCSCs were collected from GSE43793. The smRNA-seq and RNA-seq data for the TCGA COAD dataset were retrieved from established databases: DriverDB [29] and YM500 [30]. In-house pipelines were used to estimate the expression levels of miRNAs (30) as reads per million (RPM) values from fastq files. Gene expression array and microRNA array data of the NCI-60 cell line panel implemented with Affymetrix HG-U133 Plus 2 and Agilent Human microRNA-V2 chip platforms, respectively, were downloaded from the CellMiner database [31]. We used Gene Set Enrichment Analysis (GSEA) (http://www.broadinstitute.org/gsea (accessed on 20 June 2020) to assess degree of association defined signature and expression profiles of CRC patients downloaded from GSE17538. The clinical phenotypes were used for permutation.

### 2.13. Immunoblotting

Whole-cell lysates were extracted with cell culture lysis buffer (Promega, Madison, WI, USA), and protein concentrations were quantified with a Pierce BCA Protein Assay Kit (Thermo Fisher Scientific, USA) according to the manufacturer’s protocol. The transfer membrane was blocked and probed with the following antibodies prepared in 5% BSA (Sigma-Aldrich) overnight at 4 °C: anti-STMN1 (1:1000, 13655S, Cell Signaling, Danvers, MA, USA), anti-FLAG-M2-HRP (1:1000, A8592, Sigma-Aldrich), and anti-β-actin (1:5000, A5441, Sigma-Aldrich). The membrane was then probed at room temperature for 1 h with the corresponding secondary antibodies: bovine anti-rabbit IgG-HRP (1:3000, sc-2370, Santa Cruz Biotechnology, Dallas, TX, USA) and chicken anti-mouse IgG-HRP (1:5000, sc-2954, Santa Cruz Biotechnology). Immunoblots were visualized in an ImageQuant LAS 4000 chemiluminescence detection system (GE Healthcare Bio-Sciences, Piscataway, NJ, USA). All Uncropped blots can be seen in Appendix A.

### 2.14. Immunohistochemical (IHC) Assay

Sections of tissues (4 μm thick) from microarrays were deparaffinized and rehydrated before staining. Tissue antigens were retrieved by autoclaving in 10 mM (pH 6) citrate buffer for 10 min. Sections were cooled on ice for 30 min before treatment with 3% H_2_O_2_. Samples were permeabilized with 0.2% Triton X-100 (Sigma) in DPBS and reacted with a diluted primary STMN1 antibody (1:200, 13655S, Cell Signaling). Signals were amplified and detected with a Super Sensitive^TM^ Link-Label IHC Detection System (BioGenex, Fremont, CA, USA) according to the instructions and counterstained with hematoxylin QS (Vector, Burlingame, CA, USA) for 20 s. The H-scores represent the percentage of STMN1 immunoreactivity-positive regions multiplied by the STMN1 staining intensity. Images were acquired with a BX43 light microscope equipped with a DP22 CCD camera (Olympus).

### 2.15. Preparation of Patient-Derived Xenografts (PDXs)

The experimental animal procedure was approved by the Institutional Animal Care and Use Committee (IACUC) of Taipei Veterans General Hospital (2018-191). The CRC specimens were first rinsed twice with DPBS and immersed in Matrigel (Becton-Dickinson, Franklin Lakes, NJ, USA) at 37 °C. The tumors were then cut into L mm^3^ pieces and implanted subcutaneously into 4-week-old female nude mice to establish patient-derived xenografts (PDXs). The mice were sacrificed, and tumors were homogenized in TRIzol^®^ reagent (Life Technologies) and subjected to total RNA isolation.

### 2.16. Biological Samples

This study was approved by the Institutional Ethics Committee/Institutional Board of Taipei Veterans General Hospital (2016-03-001BC, 2018-11-002C). Two sets of human specimens were used. First, 2 CRC specimens (one primary tumor and one unpaired liver metastatic tumor) were collected to prepare PDXs with informed consents. Second, 11 paraffin-embedded sections from the paired primary and metastatic CRC specimens collected from the tissue biobank were subjected to IHC staining.

### 2.17. Statistical Analysis

Unless indicated otherwise, Student’s *t*-test was used to assess the significance of differences. The Pearson correlation analysis was used to analyze correlations between two factors described by continuous data. The log-rank test was used for survival analysis. The *x*^2^ test was applied for comparisons of dichotomous variables. Two-tailed *p*-values of <0.05 were considered to indicate significant differences.

## 3. Results

### 3.1. Small RNA Sequencing (smRNA-seq) Reveals Enhanced miR-210-3p Expression in CRCSCs

In an attempt to discover mechanisms regulating the motility of CSCs, we initiated this study by expanding CRCSCs from two CRC cell lines, HT29 and HCT15, using a serum-free cultivation platform, because stem-like cancer populations were enriched as cancer spheroids [32]. We found that the expanded HT29- and HCT15-CRCSCs grew as suspended colonospheres (Figure 1a) and showed increased expression of stemness genes, including NANOG, POU5F1, LGR5, CD44, and SNAI1 (Appendix A).

The resultant spheroids are referred to as sphere-derived cancer stem cells (SDCSCs) hereafter. Both HT29- and HCT15-SDCSCs exhibited higher transwell migration capacity (Figure 1b) and enhanced three-dimensional (3D) vertical invasiveness (Figure 1c) than their parental cells, and cell viability was not affected (Figure 1d). The top 500 upregulated gene signature analyzed in HT29-SDCSCs (GSE14773) was positively associated with the expression profiles of recurrent (Appendix A, upper) and late stage (Appendix A, lower) CRC patients deposited at GSE17538, suggesting the expanded SDCSCs are migrating CRCSCs.

Next, we sought to identify the primary microRNA(s) (miRNAs) responsible for CRCSC motility, because miRNA deregulation is critically involved in cancer progression [33]. Global miRNA expression patterns of HT29-SDCSCs, HCT15-SDCSCs, and their parental cells (GSE43793) were explored by small RNA sequencing (smRNA-seq). The miRNAs with log counts per million (logCPM) values of >1 and fold changes of ≥2 were selected for examination of their clinical relevance. The Venn diagram shows the nine differentially expressed miRNAs in HT29-SDCSCs and HCT15-SDCSCs (Figure 1e); four miRNAs were significantly upregulated and three were downregulated in both SDCSC datasets (Figure 1f). To explore the clinical relevance of these seven dysregulated miRNAs in CRC, we evaluated these miRNAs according to the patient survival data in the TCGA COAD dataset (*n* = 425 patients) and found that miR-210-3p (miR-210) was the only miRNA both enriched in SDCSCs and associated with poor overall survival (Figure 1g). The increased expression of miR-210 in HT29-SDCSCs (Figure 1h, left panel) and HCT15-SDCSCs (Figure 1h, right panel) was confirmed. The enhanced expression of miR-210 in stage IV tumor-derived Colo205 CRC cells (Appendix A) and liver metastatic PDX specimens (Appendix A) implies roles of miR-210 in CRCSC metastasis.

### 3.2. MiR-210 Is Required for the Migration and Invasiveness of CRCSCs

We investigated the functional roles of miR-210 in SDCSCs. In human cancers, accumulated evidence suggests that defects in asymmetric cell division (ACD) and increased symmetric cell division (SCD) of somatic stem cells expand stem cell pools and fuel tumor growth [34,35]. As stem cells tend to retain the mother strand (old) DNA template in one daughter stem cell and segregate newly synthesized DNA strands to differentiating daughter cells [36], we investigated the segregation of mother strand DNA in SDCSCs by pulse-chase BrdU labeling and paired cell assays (Appendix A). Upon knockdown of miR-210 in HT29 SDCSCs with a specific antagomir (Appendix A), predominantly symmetrical BrdU segregation (i.e., SCD) was observed (Appendix A), indicating that silencing miR-210 did not promote early differentiation of SDCSCs. The observation of the reduced sphere-forming capacity of HT29-SDCSCs with stable miR-210 knockdown (Zip-210) (Figure 2a,b) but not the corresponding HCT15-SDCSCs (Zip-210) (Figure 2a,c) also suggests limited effects of miR-210 on the self-renewal of SDCSCs. The expression of strmness markers (CD44, NANOG, and POU5F1) and differentiation markers (BMP4 and CDX2) were not altered upon silencing miR-210 in both HT29- and HCT15-SDCSCs (Appendix A). In contrast, knockdown of miR-210 markedly reduced the transwell migration capacity (Figure 2d,e) and 3D vertical invasiveness (Figure 2f,g) of both HCT15- and HT29-SDCSCs without affecting cell viability (Figure 2h).

### 3.3. MiR-210 Suppresses Stathmin1 in CRCSCs

We hypothesized that miR-210 targets a critical cytoskeletal regulator mediating cell motility and invasiveness. To identify the miR-210 target(s) responsible for these functions, we identified the overlapped downregulated genes in HT29-SDCSCs (GSE14773) with the miR-210 targets predicted by SVmicro [37] and miRtar [38] software (Figure 3a). We focused on genes associated with cell motility or tumor suppression annotated in DAVID (https://david.ncifcrf.gov/home.jsp (accessed on 4 July 2014)) and examined the expression of five putative candidates identified (LRRC2, CDKN2A, PARD3, RCBTB, and STMN1) (Figure 3b). The expression of LRRC2, RCBTB2, and STMN1 was reduced in HT29- and HCT15-SDCSCs (Figure 3c). Stathmin 1 (STMN1) expression was found to be decreased in HCT15 (Figure 3d) and HT29 (Figure 3e) cells receiving miR-210 agomiRs. The decreased expression of STMN1 protein in HT29 cells receiving miR-210 agomiRs was confirmed (Figure 3f). As STMN1 is a known miR-210 target [39], a negative association between STMN1 and miR-210 levels was observed in the NCI-60 panel (Figure 3g). Decreased expression of STMN1 at the protein level was also observed in HT29- and HCT15-SDCSCs (Figure 3h). Additionally, knockdown of miR-210 expression restored the protein expression of STMN1 in HT29- and HCT15-SDCSCs, confirming the existence of the miR-210-STMN1 axis in CRCSCs (Figure 3i).

### 3.4. Stathmin1 Expression Attenuates the Motility of CRCSCs

STMN1 is a phosphoprotein regulated by extracellular signals and can bind to α/β-tubulin to modulate microtubule dynamics [40,41]. We next evaluated whether STMN1 is involved in regulating the motility CRCSCs. We found that restoration of STMN1 expression in HT29- and HCT15-SDCSCs (Figure 4a) did not alter the viability (Figure 4b), sphere-forming capacity (Figure 4c), or clonogenicity (Figure 4d,e) of these cells. Although STMN1 was shown to regulate EMT [42], restoration of STMN1 expression decreased the transwell migration ability (Figure 4f) and invasiveness (Figure 4g,h) of both HT29- and HCT15-SDCSCs without affecting a complete EMT program (Appendix A).

In an attempt to verify the participation of the miR-210-STMN1 axis in mediating SDCSC motility, we silenced STMN1 expression in miR-210 knockdown HCT15-SDCSCs. The reduced expression of STMN1 was first confirmed (Figure 5a). We found that silencing STMN1 expression had no effects on the viability (Figure 5b), sphere-forming capacity (Figure 5c), or clonogenicity (Figure 5d,e) of miR-210-knockdown HCT15-SDCSCs. Moreover, enhanced transwell migration potential (Figure 5f) and invasiveness (Figure 5g) were noted in miR-210 knockdown HCT15-SDCSCs, but no effects on the expression of E-cadherin (CDH1), N-cadherin (CDH2) and Vimentin (VIM) were observed (Figure 5h).

### 3.5. The miR-210-STMN1 Axis Determines the Stiffness of CRCSCs

As SDCSCs must change their shapes while migrating through 8-μm pores in the transwell migration assay and through the collagen matrix in the 3D invasion assay, and the deformability of cancer cells is associated with their metastatic competence [43], we next examined the deformability of SDCSCs. To this end, we performed elasticity measurements by monitoring the time-lapse trajectory of injected fluorescent carboxylated polystyrene beads in dissociated single CRC cells (Figure 6a).

The intracellular fluorescent particles were mainly distributed in the cytoplasm (Figure 6b,c). The enhanced movement of the intracellular carboxylated polystyrene beads in HCT15-SDCSCs (Appendix A) suggested the reduced intracellular elasticity (i.e., enhanced deformability) of SDCSCs (Figure 6d, bars 1–2). Furthermore, knockdown of miR-210 (Zip-210) in HCT15-SDCSCs restored the elastic modulus (stiffness), and silencing STMN1 expression reversed the reduction in elasticity in miR-210 knockdown HCT15-SDCSCs (Figure 6d, bars 3–4). In contrast, interference with the miR-210-STMN1 axis had limited impact on the 2.5D horizontal movement of HCT15-SDCSCs on the collagen gel (Figure 6e). Collectively, these results indicated that the miR-210-STMN1 axis determined the deformability of SDCSCs to facilitate their motility.

### 3.6. The miR-210-STMN1 Axis Promotes the Stiffness of CRC Cells

To validate impacts of the miR-210-STMN1 axis on parental CRC cells, we ectopically express STMN1 in HT29 parental cells receiving miR-210 agomiRs. The expression of miR-210 (Appendix A) and Myc-DDK-tagged STMN1 (Appendix A) were conformed. It was found that the miR-210-STMN1 axis had limited impacts on the viability (Appendix A) and clonogenicity (Appendix A) of HT29 parental cells. On the contrary, the enhanced motility (Appendix A) and reduced elasticity (Appendix A) of HT29 cells receiving miR-210 agomiR could be reverted upon expressing STMN1 without modulating the EMT markers (CDH1. CDH2 and VIM) (Appendix A). Consistently, we found that expression of STMN1 in parental HT29 cells (Appendix A) had no effect on cell viability (Appendix A), clonogenicity (Appendix A) or EMT marker expression (Appendix A). However, the transwell migration ability was reduced (Appendix A) and the cellular elasticity was elevated (Appendix A) upon STMN1 expression. This results indicated the miR-210-STMN1 axis identified in CRCSCs also contributes to deformability and motility of CRC cells.

### 3.7. The miR-210^High^/STMN1^Low^ Expression Signature Is Associated with Liver Metastasis and Predicts a Poor Clinical Outcome in CRC Patients

As enhanced cellular deformability benefits local dissemination in the extracellular matrix (ECM), along with intravasation and extravasation, the miR-210^High^/STMN1^Low^ expression signature may predict distal metastasis in CRC patients. To this end, we examined the expression of miR-210-STMN1 axis components in primary and paired liver metastatic CRC specimens from our collection and databases in the public domain. Increased expression of miR-210 and decreased expression of STMN1 were observed in liver metastatic CRC samples from GSE54088 and GSE3964 datasets, respectively (Figure 6f). Additionally, decreased STMN expression in liver metastastic CRC patients was verified in paired, paraffin-embedded tissues by IHC staining (Appendix A). However, the expression of STMN1 were not changed in lymph node metastatic (Appendix A) or lung metastatic CRC specimens (Appendix A) comparing to liver metastasis of CRC. As miR-210^High^/STMN1^Low^ expression signature was associated with CRC metastasis that contributes to poor patient outcomes, we verified the clinical significance of the miR-210^High^/STMN1^Low^ expression pattern. In analysis of the TCGA data sets, the miR-210^High^/STMN1^Low^ expression signature predicted worse CRC patient survival (Figure 6g). Taken together, our results suggested the elevated expression of miR-210 attenuated STMN1 expression to engender deformability of CRCSCs for facilitating invasiveness, resulting in poor prognosis of CRC patients (Figure 6h).

## 4. Discussion

STMN1, also called oncoprotein 18 (Op18), metablastin (p19) and prosolin, is identified as a cytosolic microtubule-destabilizing phosphoprotein [44]. Unphosphorylated STMN1 promotes microtubule depolymerization by sequestering soluble tubulin and promotes microtubule catastrophe [45,46]. STMN1 contains four serine phosphorylation sites (Ser16, Ser25, Ser38, and Ser63), and the microtubule-destabilizing ability of STMN1 is regulated by its phosphorylation [47,48]. Phosphorylation of STMN1 in early mitosis abolishes its microtubule-destabilizing ability, allowing the formation of mitotic spindles, and it becomes dephosphorylated when cells exit mitosis and undergo cytokinesis [40]. Overexpression or inhibition of STMN1 expression in K562 cells resulted in accumulation of mitotic cells that were arrested in early and late mitotic stages, respectively [40,49], suggesting a threshold level of STMN1 is required for mitosis progression. Aside from cell cycle regulation, roles of STMN1 in hematopoiesis have been addressed in leukemia cells. STMN1 is abundant in acute leukemia blasts [50] and its expression was decreased when inducing differentiation by exposing an acute promyelocytic leukemia cell line HL60 to Me_2_SO or exposing erythroleukemia cells K562 to hemin [51]. Inhibition of STMN1 promoted higher megakaryocytic differentiation and polyploidization of phorbol ester-induced K562 cells [52]. On the contrary, overexpression of STMN1 in human primary CD34(+) cells reduced the megakaryocyte maturation and platelet production [53]. The megaloblastic anemia and thrombocytosis phenotypes observed in aged Stmn1 knockout mice further support STMN1′s roles in hematopoiesis [54]. Additionally, aged Stmn1 deficient mice also developed a progressive axonopathy [55]. Under social defeat stress, Stmn1 deficient mice showed anxious hyperactivity, impaired object recognition, and decreased levels of social investigating behaviors [56]. Thus, pleiotropic roles of STMN1 are highlighted.

In cancers, STMN1 expression correlates with a malignant phenotypes and has been suggested as a therapeutic target [57]. Silencing STMN1 expression inhibited the metastatic ability of a CRC cell line HCT-116 [58]. STMN1 expression was associated with aggressive phenotypes in breast cancer [59]. The oncogenic Stathmin1 is also regulated by a tumor suppressor miRNA-223 in gastric cancer [60] and liver cancer [61] or a tumor suppressor miR-34a in prostate cancer [62]. Overexpression of the somatic STMN1 Q18E mutation identified in esophageal adenocarcinoma promoted the malignant transformation of 3T3 fibroblast cells [63] and chromosomal instability in K562 cells [64]. A S31Y STMN1 missense mutation was noted in colorectal cancer patients analyzed with TumorPortal (http://www.tumorportal.org (accessed on 4 May 2020)) without functional annotation [65]. Nevertheless, D’Andrea et al. showed that Stmn1 knockout mice showed no impact on the onset of the p53-dependent nor RAS-driven tumorigenesis in bladder and fibrosarcomas or skin carcinomas in mice, respectively [66], suggesting cellular context may contribute to diverse functions of STMN1 during oncogenesis.

As local inactivation of STMN1 at the leading edge of the migrating Xenopus A6 cells potentiated localized microtubule growth, STMN1 may function as a negative regulator in cell movement [67]. Consistently, tumor suppressive roles of STMN1 were identified in prostate cancer cells [42]. Williams et al. showed that the highly invasive, EMT-like prostate cancer cells isolated from undifferentiated adenocarcinoma exhibited low STMN1 expression. Inhibition of STMN1 expression in a prostate cancer cell line DU-145, a standard prostate cancer cell line used for CSC enrichment [68], accelerated the metastatic process by initiating an EMT program via activation of p38 and cooperation of TGF-β signaling [42]. In this study, we identified an increased expression of an oncomiR miR-210 (Figure 1h) in both HT29- and HCT15-CRCSCs characterized previously [24] and showed the miR-210 mediated STMN1 suppression in CRCSCs (Figure 3i). The tumor suppressive roles of STMN1 were demonstrated by observing the reduced invasiveness of STMN1-restored CRCSCs (Figure 4g,h) and decreased motility of STMN1-overexpressed HT29 cells (Appendix A), a CRC cell line exhibits higher stem-like properties [24]. The EMT program was found to be disconnected from the miR-210-STMN1 activated invasiveness of both CRCSCs (Figure 5h) and HT29 cells (Appendix A). Here, we unraveled metastatic inhibition effects of STMN1 in our CRC cell models. Reduced STMN1 expression was also observed in paired, liver metastatic CRC specimens (Appendix A). Taken together, these findings indicate that STMN1 tends to function as a metastatic suppressor in stem-like tumor cells and suggest that understanding the stemness profiles and numbers of stem-like cells in cancer patients are crucial for utilizing STMN1 as a therapeutic target.

According to the present results and our previous findings about CRCSCs, we propose a model in which CRCSCs trigger different signaling pathways to maintain cancer stemness and subsequent metastasis: CRCSCs are Snail-dominant cells that undergo EMT [24]. In CRCSCs, Snail suppresses E-cadherin, leading to EMT and cellular disaggregation. Decreased E-cadherin expression results in nuclear translocation of β-catenin and activation of the Wnt pathway, which induces miR-146a expression in CRCSCs. During serum-induced differentiation, mIR-146a could be segregated non-randomly into CD44(+), Snail(+) daughter colorectal stem cells to initiate a feedforward β-catenin/TCF signaling to maintain stem cell pools without promoting CRCSC migration by targeting *NUMB* [35]. Here, we identified one additional oncomiR, namely miR-210, that suppressed *STMN1* expression to facilitate invasiveness of CRCSCs. The ectopic Snail expression was not found to activate the miR-210-STMN1 axis in CRC cells (Appendix A), indicating the miR-210-STMN1 axis was disconnected from an EMT program. Our findings suggest that Snail-dominant CRCSCs uncouple cancer cell division mode and deformability by utilizing distinct miRNAs for maintaining aggressive CSC phenotypes. The sequential activation of miR-210 and miR-146 and the collaboration of these miRNAs with other coding and noncoding genes in the TME require further investigation.

Our study has some limitations. First, our findings mainly rely on CRC cell line-derived cell models, primary cells or CSCs isolated from different tumor types may help to delineate dual roles of STMN1 under diverse context of cells or tissues. Second, the molecular mediators driving dual roles of STMN1 and STMN1-driven metastasis need further exploration.

## 5. Conclusions

The significance of our study is double-edged. Scientifically, the miR-210-STMN1 axis determines the invasiveness of CRCSCs without affecting cancer stemness. As the low STMN1 expression is essential for migrating of CRCSCs, using STMN1 as a therapeutic target might accelerate metastasis of CRCSCs. Clinically, this study proposes a miR-210^High^/STMN1^Low^ expression pattern as a potential indicator for monitoring the liver metastasis longitudinally along with therapeutic regimes.

## Figures and Tables

**Figure 1 cancers-13-01833-f001:**
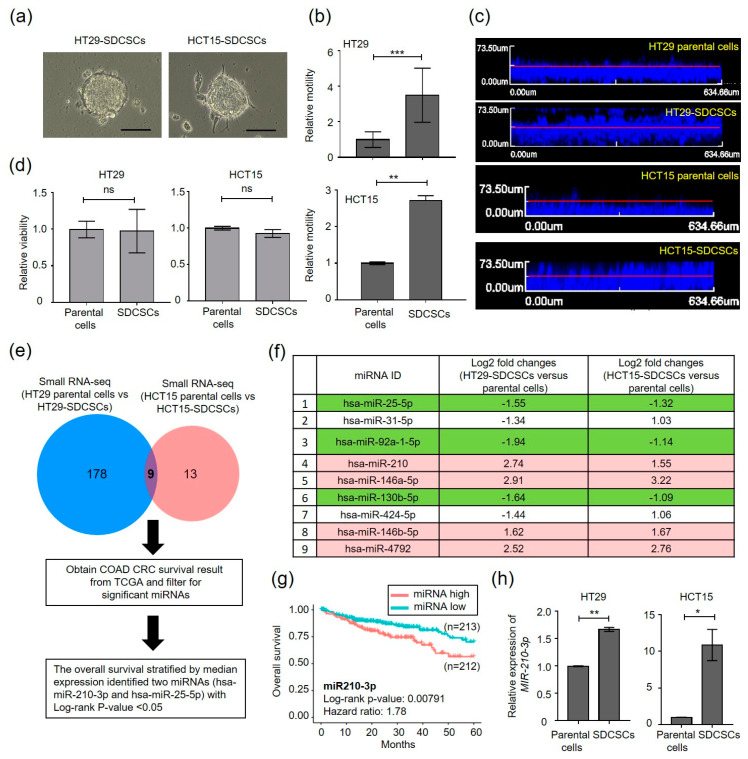
Increased expression of miR-210-3p in SDCSCs. (**a**) Representative images of SDCSCs from two CRC cell lines. SDCSCs, sphere-derived cancer stem cells. Scale bar, 100 μm. (**b**) Histograms showing the relative transwell migration ability of cells. (**c**) Representative images of vertical invasion of cells. (**d**) Relative viability of cells as assessed by an MTT assay. ns, nonsignificant. (**e**) Flow charts for identifying differentially expressed miRNAs in the CRCSC miRNome associated with poor patient outcome in the TCGA-COAD dataset. The numbers of miRNAs with the same differential regulation patterns in HT29- and HCT15-SDCSCs are indicated in Venn diagrams. TCGA-COAD, The Cancer Genome Atlas Colon Adenocarcinoma. (**f**) A table illustrating the relative miRNA expression levels in SDCSCs. Red, miRNAs upregulated in both SDCSC lines. Green, miRNAs downregulated in both SDCSC lines. (**g**) Kaplan–Meier analysis of overall survival in patients in a TCGA-COAD dataset (*n* = 425). The median miR-210-3p expression level was used for patient stratification. miRNA high, CRC patients with high miR-210-3p expression. miRNA low, patients with low miR-210-3p expression. The *p*-value was estimated by the log-rank test. (**h**) RT-qPCR validation of miR-210-3p expression in the indicated cells. Unless otherwise stated, all data in bar charts are expressed as the mean ± SD values. * *p* < 0.05; ** *p* < 0.01; *** *p* < 0.001 (Student’s *t*-test).

**Figure 2 cancers-13-01833-f002:**
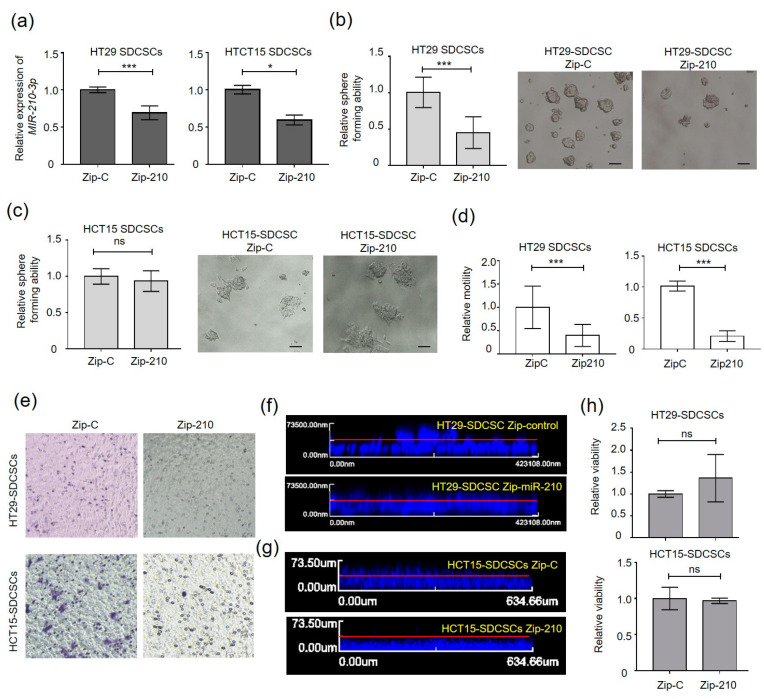
Silencing miR-210-3p expression inhibits the migratory and invasive capabilities of SDCSCs. (**a**) RT-qPCR validation of miR-210-3p expression in SDCSCs. Zip-C, cells receiving the scramble control; Zip-210, miR-210-3p-knockdown cells receiving a shRNA targeting miR-210-3p. (**b**) Left: sphere-forming capacity of HT29-SDCSCs receiving the scramble control (Zip-C) or shRNA targeting miR-210-3p (Zip-210). Right: representative pictures of sphere formation in miR-210-3p-knockdown SDCSCs. Scale bar, 50 μm. (**c**) Left: sphere-forming capacity of control HCT15-SDCSCs (Zip-C) and miR-210-3p-knockdown SDCSCs (Zip-210). Right: representative pictures of sphere formation in miR-210-3p-knockdown SDCSCs. Scale bar, 50 μm. (**d**) Histograms showing the relative transwell migration ability of cells. (**e**) Representative images showing migrated SDCSCs. (**f**) Representative images of vertical invasion of control HT29-SDCSCs and miR-210-3p-silenced HT29-SDCSCs. (**g**) Representative images of vertical invasion of control HCT15-SDCSCs and miR-210-3p-silenced HCT15-SDCSCs. (**h**) Relative viability of cells as assessed by an MTT assay. ns, nonsignificant. All data in bar charts are expressed as the mean ± SD values. * *p* < 0.05; *** *p* < 0.001 (Student’s *t*-test).

**Figure 3 cancers-13-01833-f003:**
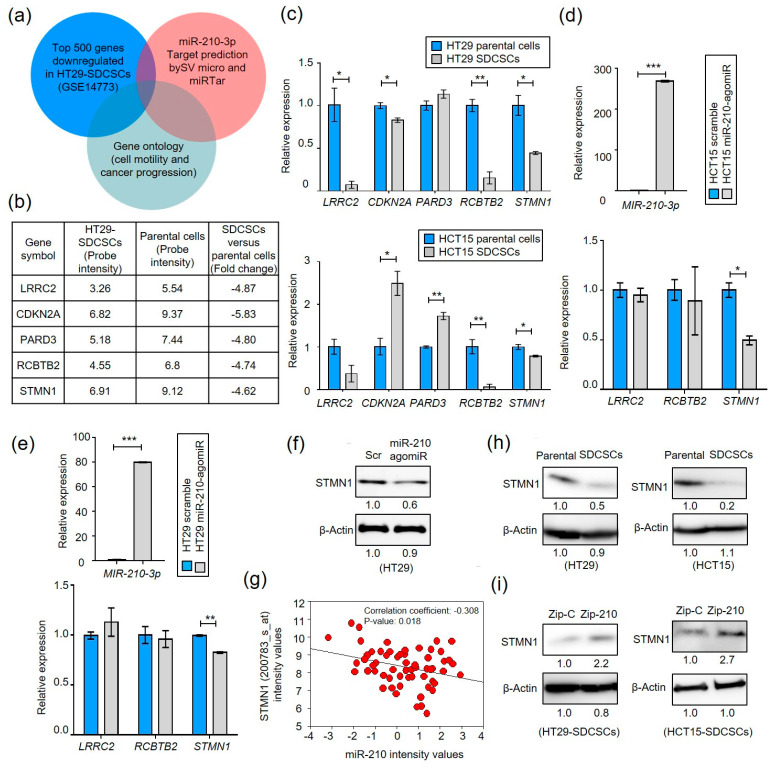
Identification of the miR-210-3p-STMN1 axis in SDCSCs. (**a**) Strategy for identifying miR-210 targets. The putative targets were obtained from the set of differentially expressed genes in SDCSCs versus parental HT29 cells (GSE14773) and subjected to target prediction with SVmicro. Migration-, tumor suppressor- and stemness-related genes were selected. (**b**) A table illustrating the expression levels of five candidate targets of miR-210-3p from microarray analysis. (**c**) RT-qPCR examining the expression of five candidate targets of miR-210-3p in SDCSCs and their parental cells. (**d**,**e**) RT-qPCR examining the expression of miR-210-3p or LRRC2, RCBTB2 and STMN1 in HCT15 cells (**d**) and HT29 cells transfected with control agomiR (Scramble, 100 nM) or miR-210-3p agomiR (miR-210-AgomiR, 100 nM). (**f**) Western blot showing the expression of STMN1 in HT29 cells transfected with control agomiR (Scramble, Scr, 100 nM) or miR-210-3p agomiR (miR-210-AgomiR, 100nM). (**g**) Analysis of the NCI-60 panel revealed an inverse correlation between STMN1 (200783_s_at) and miR-210-3p expression. The pvalue of the Pearson correlation was assessed, and the correlation coefficient is reported. (**h**) Immunoblots showing STMN1 expression. (**i**) Western blots showing the expression of STMN1 in control SDCSCs (Zip-C) and miR-210-3p-silenced SDCSCs (Zip-210). Data in bar charts are expressed as the mean ± SD values. * *p* < 0.05; ** *p* < 0.01; *** *p* < 0.001 (Student’s *t*-test).

**Figure 4 cancers-13-01833-f004:**
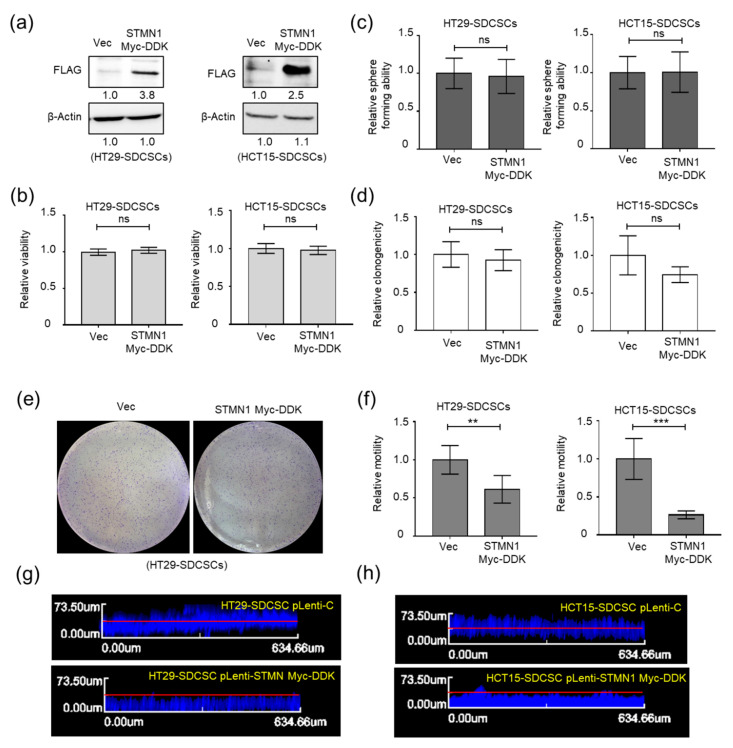
Restoration of STMN1 expression abolishes the migratory and invasive abilities of SDCSCs. (**a**) Western blots showing the expression of STMN1 in control SDCSCs (Vec) and SDCSCs ectopically expressing STMN1. An anti-FLAG antibody was used to detect the expression of exogenous STMN1 with a DKK-Myc tag. (**b**) Relative viability of cells as assessed by an MTT assay. ns, nonsignificant. (**c**) The sphere-forming capacity of control SDCSCs (Vec) and STMN-expressing SDCSCs (STMN1 DKK-Myc). ns, nonsignificant. (**d**) The colony formation of control SDCSCs (Vec) and STMN-expressing SDCSCs (STMN1 DKK-Myc). ns, nonsignificant. (**e**) Representative images showing the colonies generated. (**f**) Histograms showing the relative transwell migration ability of cells. (**g**,**h**) Representative images of vertical invasion of vector control HT29-SDCSCs (pLenti-C) and STMN-expressing SDCSCs (pLenti-STMN1 Myc-DDK). Data in bar charts are expressed as the mean ± SD values. ** *p* < 0.01; *** *p* < 0.001 (Student’s *t*-test).

**Figure 5 cancers-13-01833-f005:**
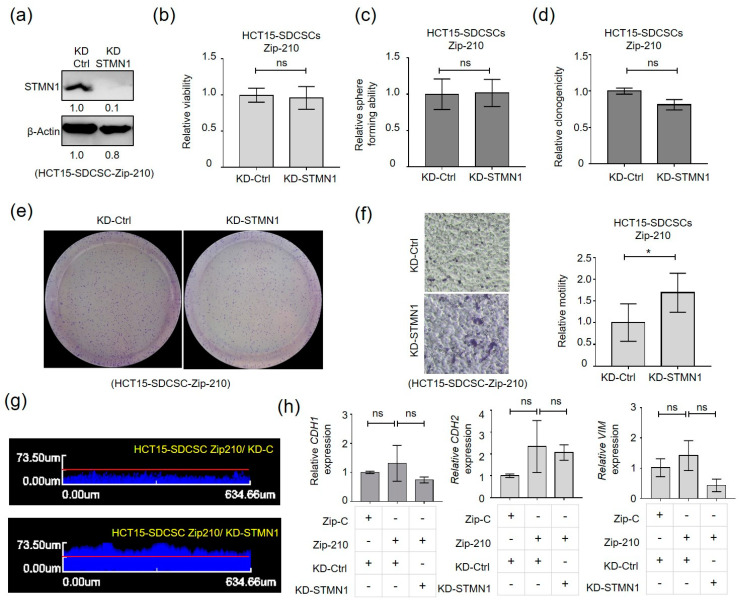
Silencing STMN1 expression restored the attenuated migratory and invasive abilities of miR-210-3p-knockdown SDCSCs. (**a**) Western blots showing the expression of STMN1 in miR-210-3p-knockdown HCT15-SDCSCs. (**b**) Relative viability of cells as assessed by an MTT assay. ns, nonsignificant. (**c**) The sphere-forming capacity of miR-210-3p-silenced HCT15-SDCSCs receiving control shRNA (KD-Ctrl) and shRNA against STMN1 (KD-STMN1). ns, nonsignificant. (**d**) The colony formation ability of miR-210-3p-silenced HCT15-SDCSCs receiving control shRNA (KD-Ctrl) and shRNA against STMN1 (KD-STMN1). ns, nonsignificant. (**e**) Representative images showing colonies generated from the indicated cells. (**f**) Left: representative images of migrated cells. Right: histograms showing the relative transwell migration ability of cells. (**g**) Representative images of vertical invasion of the indicated cells. (**h**) RT-qPCR validation of the expression of an epithelial cell marker (E-cadherin, CDH1) and mesenchymal marker (N-cadherin, CDH2) in control HCT15-SDCSCs, miR-210-3p-silenced HCT15-SDCSCs receiving shRNA control (KD-Ctrl) and miR-210-3p-silenced HCT15-SDCSCs receiving shRNA against STMN1 (KD-STMN1). ns, nonsignificant. Data in bar charts are expressed as the mean ± SD values. * *p* < 0.05 (Student’s *t*-test).

**Figure 6 cancers-13-01833-f006:**
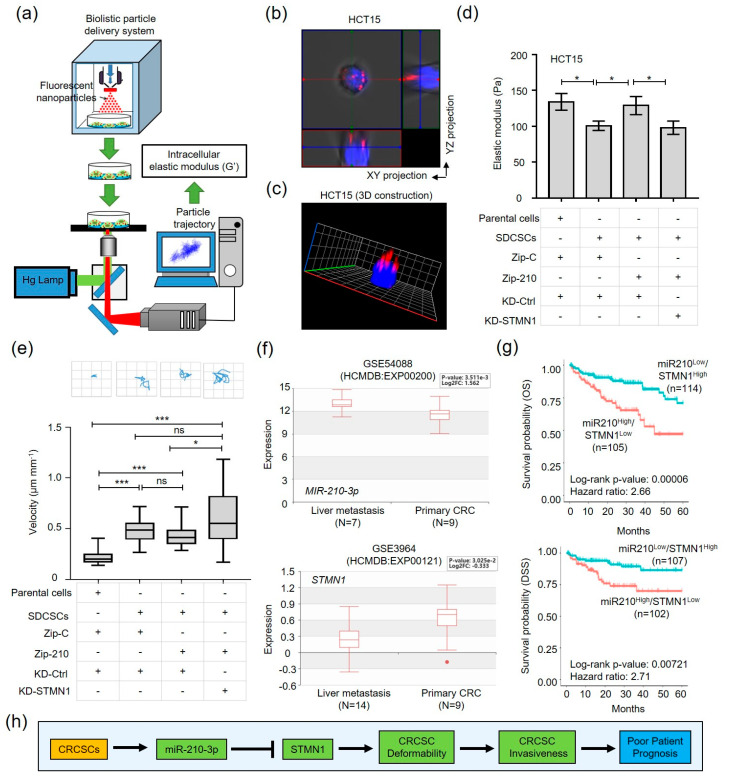
The miR-210-3p-STMN1 axis determines the elasticity of SDCSCs and correlates with liver metastasis in CRC patients. (**a**) A representative image illustrating the biolistic particle delivery system and measurement of the intracellular elastic modulus. (**b**) Representative images showing the distribution of injected fluorescent nanoparticles in HCT15 cells by confocal imaging. The XY and YZ projections are shown. Blue, nuclear staining with Hoechst 33342; red, fluorescent nanoparticles. (**c**) A three-dimensional (3D) constructed image from (**b**). (**d**) Histograms showing the elastic modulus of the indicated HCT15 cells at 100 Hz. Zip-C, scramble control; Zip-210, shRNA against miR-210-3p; KD-Ctrl, scramble control; KD-STMN1, shRNA targeting STMN1. Results are expressed as the mean ± SEM values. (**e**) Upper: trajectory of the indicated cells in the 2.5D assay. Lower: the velocity values in the indicated cells. The box plots show the sample maximum (upper end of the whisker), upper quartile (top edge of the box), median (band in the box), lower quartile (bottom edge of the box), and sample minimum (lower end of the whisker) values. * *p* < 0.05; *** *p* < 0.001 (Student’s *t*-test). (**f**) The bar charts showing the expression of miR-210-3p (upper panel) and STMN1 (lower panel) in GSE54088 and GSE3964 datasets retrieved from the Human Cancer Metastasis Database (HCMDB), respectively. The box plots show the sample maximum (upper end of the whisker), upper quartile (top edge of the box), median (band in the box), lower quartile (bottom edge of the box), and sample minimum (lower end of the whisker) values. (**g**) Kaplan–Meier analysis of the overall survival (upper panel) and disease-specific survival (lower panel) of patients in a TCGA-COAD dataset. The median expression levels of miR-210-3p and STMN1 were used for patient stratification. OS, overall survival; DSS, disease-specific survival. The *p*-values were estimated by the log-rank test. Hazard ratios are reported. (**h**) A schematic summarizing the observations in this study.

## Data Availability

The datasets used and analyzed during the current study are available from the corresponding author on reasonable request.

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
