# Peer review of "The microRNA-210-Stathmin1 Axis Decreases Cell Stiffness to Facilitate the Invasiveness of Colorectal Cancer Stem Cells"

_cancers, 2021, doi:10.3390/cancers13081833_

Round 1

Reviewer 1 Report

The MS by Liao et al. reported the importance of mircoRNA-210-STMN1 axis in maintaining cell motility and metastasis of colorectal cancer stem cells using several biological techniques. The authors firstly found that smRNA-Seq revealed an up-regulated miR-210-3p in CRCSCs. The authors then showed the functional requirement of MiR-210 for the migration and invasiveness of CRCSCs. The authors finally showed the connection between MiR-210 and Stathmin1 in CRCSCs with the functional importance of this connection. The flow of the story is really well. However, there are some weaknesses that need to be improved to enhance the quality of this manuscript that will be suitable for publication in this journal

 Minor comments

  1. The authors are suggested to look at the cancer stem cell markers in CRCSCs-derived from both of their cell lines.
  2. In line 489 and 490 it is written that “Additionally, decreased STMN expression was verified in paraffin- 489 embedded tissues by IHC staining (Figure. 5c and 5d)”. But there are no IHC staining in figure 5C and 5D. The authors are strongly suggested to include this data in their main figures.
  3. What are the phenotypes associated with Stath1 deficient mice? The authors can discuss this point in their discussion section.
  4. It has been known that mRNA-210 regulates apoptosis. Are there any association between different forms of cell death such as Pyroptosis, Apoptosis, Necroptosis (PANoptosis) and STMN1 activation?
  5. Are there any reported mutations with STMN1 in colorectal patients? Discuss this in your discussion section.
  6. The supplementary data are not available even via online link.
  7. Graphical abstract need to be corrected. Please include only the molecules that you found in this study in your graphical abstract.

Major comments

  1. The authors are suggested to experimentally show how mRNA-210 controls STMN1?
  2. Have the authors tested the importance of this axis in other cancer cell lines? Is this phenotype specific to colorectal cancer? If yes, please show at least in some other cell lines as well before making this conclusion.
  3. The authors are suggested to check differentiation, proliferation, and EMT markers before excluding the roles of this process in this phenotype.
  4. The points mentioned in discussion are very poor. The authors are suggested to improve their discussion. Please discuss/compare your findings with similar findings in literatures; discuss your unusual findings; discuss your limitations; recommend future directions; discuss the physiological relevance.

Author Response

Please find the attaching PDF file for the authors' reply. Thank you.

Minor comments
1. The authors are suggested to look at the cancer stem cell markers in CRCSCs-derived from both of their cell lines.
Authors’ reply: The CRCSC expansion platform has been established and
characterized previously [1]. In this study, we expanded CRCSCs using this platform again, and the increased expression of cancer stem cell-related transcriptional factors (NANOG, OCT4, and SNAI1) and cancer stem cell markers (CD44 and LGR5) was increased in both HT29- andHCT15-SDCSCs (Figure. S1a). The top 500 up-regulated signatures of HT29-SDCSCs further correlated with expression profiles of recurrent and late-stage CRC patients (Figure. S1b), suggesting the clinical relevance of expanded CRCSCs.
2. In line 489 and 490 it is written that “Additionally, decreased STMN expression was verified in paraffin- 489 embedded tissues by IHC staining (Figure. 5c and 5d)”. But there are no IHC staining in figure 5C and 5D. The authors are strongly suggested to include this data in their main figures.
Authors’ reply: The results were originally shown in supplementary figures 5c and 5d and now in supplementary figures 6a and 6b. We place these results as supporting results because of a lack of examining miR-210 expression profile in paired CRC specimens due to the limitation of sample collection. The inversed expression pattern of the miR-210-STMN1 axis in primary and liver metastasized CRC specimens analyzed from the Human Cancer Molecular Database (HCMDB) was shown in figure 6f. Thank you for this suggestion.
3. What are the phenotypes associated with Stath1 deficient mice? The authors can discuss this point in their discussion section.
Authors’ reply: Stmn1 deficient mice showed anxious hyperactivity, impaired object recognition, and decreased levels of social investigating behaviors under social defeat stress [2]. The megaloblastic anemia and thrombocytosis phenotypes were observed in aged Stmn1 knockout mice [3]. Aged Stmn1 deficient mice also developed a progressive axonopathy [4]. The above phenotypes associated with Stmn1 deficient mice have been incorporated into the discussion (Line 544-548). Thank you for this comment.
4. It has been known that mRNA-210 regulates apoptosis. Are there any association between different forms of cell death such as Pyroptosis, Apoptosis, Necroptosis (PANoptosis) and STMN1 activation?
Authors’ reply: STMN1 activation has been known to regulate apoptosis. The
apoptosis was followed by the induction of hepatocyte proliferation by Stmn1
overexpression in mice [5]. Overexpression of constitutively active STMN1 promoted apoptosis and reduced cell proliferation in HCV-replicon harboring HCC cells (RHuh7.5) [6]. We found the cellular viability was not altered when we knocked down miR-210 expression (Figure. 2h), overexpressed STMN1 (Figure. 4b) or manipulated the miR-210-STMN1 axis (Figure. 5b) in SDCSCs. The miR-210-STMN1 axis is less likely to associate with cell death in our model cells.
5. Are there any reported mutations with STMN1 in colorectal patients? Discuss this in your discussion section.
Authors’ reply: We did not identify any reported mutations with STMN1 in CRC
patients. To confirm this point, we survey the somatic point mutations of STMN1 in 4,742 human cancers and their matched normal-tissue samples across 21 cancer types, including colorectal cancer by TumorPortal [7]. The S31Y STMN1 missense mutation was noted in colorectal cancer patients, indicated by the upper arrow above. The functionality of this S31Y STMN1 mutation needs further characterization. The Q18E mutation identified in esophageal adenocarcinoma [8] is also reported in TumorPortal. We also incorporate related descriptions in the discussion (Line 555-559).
6. The supplementary data are not available even via online link.
Authors’ reply: Thank you for this notification, and we will confirm the availability
of supplementary results and all the other documents with the Editorial office to
facilitate the review process.
7. Graphical abstract need to be corrected. Please include only the molecules that you found in this study in your graphical abstract.
Authors’ reply: We have modified the graphical abstract as showing in figure 6h.
Thank you for this suggestion.
Major comments
8. The authors are suggested to experimentally show how mRNA-210 controls STMN1?
Authors’ reply: The miR-210-STMN1 interaction has been addressed during virus
infection [9]. The miR-210 expression was epigenetically silenced during chronic
Helicobacter pylori infection and inhibited gastric cell proliferation by suppressing STMN1. The miR-210 was bound to 3’UTR of STMN1 by its seeding regions, and the direct miRNA-STMN1 interaction was verified using an elegant 3’UTR reporter assay with mutant miR-210 and mutant 3’UTR reporter constructs. Therefore, we then examined the miR-210-STMN1 axis through manipulating miR-210 expression in parental CRC cells (Figure. 3d-f) and SDCSCs (Figure. 3i). Next, the inverse expression pattern of the miR-210-STMN1 axis was validated in the NCI-60 cell panel (Figure. 3g) and liver metastasized CRC specimens (Figure. 6f). More functional assays were employed in this study to dissect the significance of the miR-210-STMN1 axis in CRCSCs.
9. Have the authors tested the importance of this axis in other cancer cell lines? Is this phenotype specific to colorectal cancer? If yes, please show at least in some other cell lines as well before making this conclusion.
Authors’ reply: We currently investigated the significance of the miR-210-STMN1
axis primarily in CRCSCs. To verify the biological relevance of the miR-210-STMN1 axis in the CRC cell line, we transfected the miR-210 agomiRs in HT29 cells and restored the STMN1 expression for functional assays. Consistently, the miR-210- STMN1 axis regulated the elasticity of HT29 cells (Figure. S4g). Overexpression of STMN1 also enhanced the elasticity of HT29 cells (Figure. S5g). We have now emphasized the impacts of the miR-210-STMN1 axis and STMN1 itself in stem-like cancer cells, as discussed in lines 566-585.
10. The authors are suggested to check differentiation, proliferation, and EMT markers before excluding the roles of this process in this phenotype.
Authors’ reply: Thank you for the suggestion. First, we conducted the paired-cell
assay (Figure. S2a-d), spheroid formation assay (Figure. 2b-c) and examined the
expression of stemness markers including CD44, NANOG, and OCT4, and
differentiation markers such as BMP4 and CDX2 (Figure S2e) in miR-210 silenced
SDCSCs. We found knocking down miR-210 in SDCSCs had limited effects on selfrenewal and differentiation. Second, expression of the EMT markers (CDH1, CDH2, and VIM) was not consistent throughout STMN1-restored SDCSCs (Figure. S3), STMN1-overexpressed HT29 cells (Figure. S5e), miR-210-STMN1 axis manipulated SDCSCs (Figure. 5h), and miR-210-STMN1 axis manipulated HT29 cells (Figure.S4h). Third, the viability was not altered when we knocked down miR-210 expression (Figure. 2h), overexpressed STMN1 (Figure. 4b) in SDCSCs, or miR-210-STMN1 axis manipulated SDCSCs (Figure. 5b). The unchanged viability was represented in STMN1-overexpressed (Figure. S5b) and miR-210-STMN1 axis manipulated HT29 cells (Figure. S4c). On the basis of the findings above, the miR-210-STMN1 axis may not contribute to the differentiation, proliferation, and EMT status in our CRC model
cells. The underlines are new results provided in this revision.
11.The points mentioned in discussion are very poor. The authors are suggested to improve their discussion. Please discuss/compare your findings with similar findings in literatures; discuss your unusual findings; discuss your limitations; recommend future directions; discuss the physiological relevance.
Authors’ reply: Thank you for this precious suggestion. We now reorganize and
incorporate the phenotypes of Stmn1 deficient mice and STMN1 mutation of CRC patients into the discussion section. The physiological relevance of STMN1 targeting oncomiR miR-210, and tumor suppressor miRNAs including miR-223 [10] and miR-34a [11] as well as dual roles of STMN1 in cancers, were highlighted in the discussion to recommend therapeutic direction. The limitation of this study is also described in the discussion. Please find the discussion section for details.
1. Hwang WL, Yang MH, Tsai ML, Lan HY, Su SH, Chang SC, et al. SNAIL regulates interleukin-8 expression, stem cell-like activity, and tumorigenicity of human colorectal carcinoma cells. Gastroenterology 2011;141(1):279-91, 91 e1-5 doi 10.1053/j.gastro.2011.04.008.
2. Nguyen TB, Prabhu VV, Piao YH, Oh YE, Zahra RF, Chung YC. Effects of Stathmin 1 Gene Knockout on Behaviors and Dopaminergic Markers in Mice Exposed to Social Defeat Stress. BrainSci 2019;9(9) doi 10.3390/brainsci9090215.
3. Ramlogan-Steel C. A., Steel J. C., Fathallah H., Iancu-Rubin C., Soleimani M., Dong Z., Atweh G. F. The role of Stathmin, a regulator of mitosis, in hematopoiesis. Blood (ASH Annual Meeting Abstracts). (2012);120 Abstract #3453.
4. Liedtke W, Leman EE, Fyffe RE, Raine CS, Schubart UK. Stathmin-deficient mice develop an agedependent axonopathy of the central and peripheral nervous systems. Am J Pathol 2002;160(2):469-80 doi 10.1016/S0002-9440(10)64866-3.
5. Zhao E, Shen Y, Amir M, Farris AB, Czaja MJ. Stathmin 1 Induces Murine Hepatocyte Proliferation and Increased Liver Mass. Hepatol Commun 2020;4(1):38-49 doi 10.1002/hep4.1447.
6. 49. Lu NT, Liu NM, Patel D, Vu JQ, Liu L, Kim CY, et al. Oncoprotein Stathmin Modulates Sensitivity to Apoptosis in Hepatocellular Carcinoma Cells During Hepatitis C Viral Replication.J Cell Death 2018;11:1179066018785141 doi 10.1177/1179066018785141.
7. Lawrence MS, Stojanov P, Mermel CH, Robinson JT, Garraway LA, Golub TR, et al. Discovery and saturation analysis of cancer genes across 21 tumour types. Nature 2014;505(7484):495-501 doi 10.1038/nature12912.
8. Misek DE, Chang CL, Kuick R, Hinderer R, Giordano TJ, Beer DG, et al. Transforming properties of a Q18-->E mutation of the microtubule regulator Op18. Cancer Cell 2002;2(3):217-28 doi 10.1016/s1535-6108(02)00124-1.
9. Kiga K, Mimuro H, Suzuki M, Shinozaki-Ushiku A, Kobayashi T, Sanada T, et al. Epigenetic silencing of miR-210 increases the proliferation of gastric epithelium during chronic Helicobacter pylori infection. Nat Commun 2014;5:4497 doi 10.1038/ncomms5497.
10. Wong QW, Lung RW, Law PT, Lai PB, Chan KY, To KF, et al. MicroRNA-223 is commonly repressed in hepatocellular carcinoma and potentiates expression of Stathmin1. Gastroenterology 2008;135(1):257-69 doi 10.1053/j.gastro.2008.04.003.
11. Chakravarthi B, Chandrashekar DS, Agarwal S, Balasubramanya SAH, Pathi SS, Goswami MT, et al. miR-34a Regulates Expression of the Stathmin-1 Oncoprotein and Prostate Cancer Progression.
Mol Cancer Res 2018;16(7):1125-37 doi 10.1158/1541-7786.MCR-17-0230.

Reviewer 2 Report

In the manuscript titled ´The microRNA-210-Stathmin1 Axis Decreases Cell Stiffness to Facilitate the Invasiveness of Colorectal Cancer Stem Cells”, the authors reported that microRNA-210 mediated inhibition of stathmin expression in colorectal cancer stem cells resulted in cell deformity and increase invasiveness. The manuscript is well written and would be an important addition to the journal.

Author Response

Reviewer 2 Comments and Suggestions for Authors
In the manuscript titled ́The microRNA-210-Stathmin1 Axis Decreases Cell Stiffness to Facilitate the Invasiveness of Colorectal Cancer Stem Cells”, the authors reported that microRNA-210 mediated inhibition of stathmin expression in colorectal cancer stem cells resulted in cell deformity and increase invasiveness. The manuscript is well written and would be an important addition to the journal.

Authors’ reply: Thank you for your comments, and we are glad to input the scientific merit in cancer stem cell biology.

Reviewer 3 Report

This is an interesting study where the authors have identified that miR-210 was upregulated in colorectal stem-like cancer cells, which targeted stathmin1 to decrease cell elasticity and increase cell motility.

The study is well performed, and the manuscript well written. The results and discussion however do not clearly explain the role of Stathmin 1 vis-à-vis its known or reported role as a key oncoprotein.

For example, Stathmin 1 is shown to promote oncogenesis, including in colorectal (Mol Cancer Res 2014 Dec;12(12):1717-28), breast cancer (Int J Oncol 2017 Sep;51(3):781-790), gastric cancer (Br J Cancer 2017 Apr 25;116(9):1177-11850), and many other cancers. How do the authors reconcile the oncogenic role of Stathmin 1 with the anti-motility and tumor suppressor role identified in their studies?

Also, other miRNAs have been implicated in regulating Stathmin 1. For example, Stathmin1 is a known target of miRNA-223 in gastric (PLoS One. 2012;7(3):e33919) and liver (Gastroenterology. 2008 Jul;135(1):257-69) cancers. Similarly, miR-34a regulates expression of the Stathmin-1 in prostate cancer (Mol Cancer Res 2018 Jul;16(7):1125-1137). The authors need to discuss the relevance and importance of identifying miR-210 and compare their findings with the other miRNAs known to target Stathmin 1.

Author Response

This is an interesting study where the authors have identified that miR-210 was upregulated in colorectal stem-like cancer cells, which targeted stathmin1 to decrease cell elasticity and increase cell motility.
The study is well performed, and the manuscript well written. The results and
discussion however do not clearly explain the role of Stathmin 1 vis-à-vis its known or reported role as a key oncoprotein.
For example, Stathmin 1 is shown to promote oncogenesis, including in colorectal (Mol Cancer Res 2014 Dec;12(12):1717-28), breast cancer (Int J Oncol 2017 Sep;51(3):781-790), gastric cancer (Br J Cancer 2017 Apr 25;116(9):1177-11850), and many other cancers. How do the authors reconcile the oncogenic role of Stathmin 1 with the antimotility and tumor suppressor role identified in their studies?
Authors’ reply: Stathmin1 is a pleiotropic regulator involved in cell cycle regulation, differentiation, nervous system, and cognition functions [1-4]. Though Stathmin is considered initially as a therapeutic target, some studies also show no impact of STMN1 on the onset of oncogenesis and STMN1 could function as a metastasis suppressor. D’Andrea et al. showed that Stmn1 knockout mice showed no impact on the onset of the p53-dependent nor RAS-driven tumorigenesis in bladder and fibrosarcomas or skin
carcinomas in mice, respectively [5]. Williams et al. showed that the highly invasive, EMT-like prostate cancer cells could only be isolated from undifferentiated adenocarcinoma and exhibited low STMN1 expression. Inhibition of STMN1 expression in a prostate cancer cell line DU-145, which is a standard prostate cancer cell line used for CSC enrichment [6], further accelerated the metastatic process by initiating an EMT program via activation of p38 and cooperation of TGF-β signaling. In this study, the reduced expression of STMN1 was detected in colorectal cancer stem cells (CRCSCs), and restoration of STMN1 impaired the invasiveness of CRCSCs. These findings suggest STMN1 may function as a tumor suppressor predominantly in the stem-like cancer population while as an oncogene in non-CSCs. Thank you for the
comments, and the multifaceted roles of STMN1 have now been described in the discussion section (Line 542-585).

Also, other miRNAs have been implicated in regulating Stathmin 1. For example, Stathmin1 is a known target of miRNA-223 in gastric (PLoS One. 2012;7(3):e33919) and liver (Gastroenterology. 2008 Jul;135(1):257-69) cancers. Similarly, miR-34a regulates expression of the Stathmin-1 in prostate cancer (Mol Cancer Res 2018 Jul;16(7):1125-1137). The authors need to discuss the relevance and importance of identifying miR-210 and compare their findings with the other miRNAs known to target Stathmin 1

Authors’ reply: The microRNAs would function as oncomiRs or tumor suppressors by regulating the expression of their targets during cancer progression. We previously showed that miR-146a could segregate into CD44(+) daughter colorectal stem cells to initiate a feedforward β-catenin/TCF signaling to maintain stem cell pools by targeting NUMB in FBS-induced differentiation without affecting the motility of CRCSCs [7]. Here, we identified one additional miRNA, namely miR-210, that was required for invasiveness of CRCSC by targeting STMN1. Expression of both miR-146a and miR-210 ultimately contributes to aggressive phenotypes of CRCSCs. The significance of miR-210 and dual roles of the miRNA-STMN1 axis have been incorporated into the discussion section (Line 553-555, Line 572-574, and Line 589-600). The limitation of this study is also mentioned in the discussion part (Line 603-607). Thank you for these valuable suggestions.
1. Belmont LD, Mitchison TJ. Identification of a protein that interacts with tubulin dimers and increases the catastrophe rate of microtubules. Cell 1996;84(4):623-31 doi 10.1016/s0092-8674(00)81037-5.
2. Iancu-Rubin C, Gajzer D, Tripodi J, Najfeld V, Gordon RE, Hoffman R, et al. Down-regulation of stathmin expression is required for megakaryocyte maturation and platelet production. Blood 2011;117(17):4580-9 doi 10.1182/blood-2010-09-305540.
3. Liedtke W, Leman EE, Fyffe RE, Raine CS, Schubart UK. Stathmin-deficient mice develop an agedependent axonopathy of the central and peripheral nervous systems. Am J Pathol 2002;160(2):469-80 doi 10.1016/S0002-9440(10)64866-3.
4. Nguyen TB, Prabhu VV, Piao YH, Oh YE, Zahra RF, Chung YC. Effects of Stathmin 1 Gene Knockout on Behaviors and Dopaminergic Markers in Mice Exposed to Social Defeat Stress. Brain Sci 2019;9(9) doi 10.3390/brainsci9090215.
5. D'Andrea S, Berton S, Segatto I, Fabris L, Canzonieri V, Colombatti A, et al. Stathmin is dispensable for tumor onset in mice. PLoS One 2012;7(9):e45561 doi
10.1371/journal.pone.0045561.
6. Wang L, Huang X, Zheng X, Wang X, Li S, Zhang L, et al. Enrichment of prostate cancer stemlike cells from human prostate cancer cell lines by culture in serum-free medium and chemoradiotherapy. Int J Biol Sci 2013;9(5):472-9 doi 10.7150/ijbs.5855.
7. Hwang WL, Jiang JK, Yang SH, Huang TS, Lan HY, Teng HW, et al. MicroRNA-146a directs the symmetric division of Snail-dominant colorectal cancer stem cells. Nat Cell Biol 2014;16(3):268-80 doi 10.1038/ncb2910.

Round 2

Reviewer 1 Report

My comments are now addressed. I hope the authors have incorporated all these points in their revised form.